# Exploiting the speckle-correlation scattering matrix for a compact reference-free holographic image sensor

KyeoReh Lee[1] & YongKeun Park[1]

The word 'holography' means a drawing that contains all of the information for light—both amplitude and wavefront. However, because of the insufficient bandwidth of current electronics, the direct measurement of the wavefront of light has not yet been achieved. Though reference-field-assisted interferometric methods have been utilized in numerous applications, introducing a reference field raises several fundamental and practical issues. Here we demonstrate a reference-free holographic image sensor. To achieve this, we propose a speckle-correlation scattering matrix approach; light-field information passing through a thin disordered layer is recorded and retrieved from a single-shot recording of speckle intensity patterns. Self-interference via diffusive scattering enables access to impinging light-field information, when light transport in the diffusive layer is precisely calibrated. As a proof-of-concept, we demonstrate direct holographic measurements of three-dimensional optical fields using a compact device consisting of a regular image sensor and a diffusor.

---

[1] Department of Physics and KAIST Institute of Health Science and Technology, Korea Advanced Institute of Science and Technology, Daejeon 34141, Republic of Korea. Correspondence and requests for materials should be addressed to Y.P. (email: yk.park@kaist.ac.kr)

Photography does not present volumetric and depth perceptions because wavefront (or phase) information of light cannot be recorded on a film or an image sensor (Fig. 1a). The loss of wavefront information, the so-called *phase problem*[1], has been a classical issue in numerous fields, including holography[2,3], X-ray crystallography[4] and three-dimensional imaging[5].

To remedy the issue, Gabor proposed a holographic method that exploited the interferometric nature of waves[2,3]. By recording interference patterns between an unknown incident field and a known reference field, it is possible to deduce the incident field information (Fig. 1b). Due to its powerful ability, reference-assisted holography has been utilized in numerous disciplines[6–8]. Unfortunately, despite the significant efforts steadily continued for decades, the realizations and applications of holography are limited.

One prime reason is the practical issues of introducing a reference field. In an X-ray regime, for example, introducing a reference field is impractical due to the limited capability of X-ray optics[9,10]. Even in a visible regime, introducing a reference arm results in various limitations because it makes the system complex, incompatible and vulnerable to ambient noise[11]. However, although several reference-free holographic methods have been proposed, including Shack-Hartmann-type sensors[12,13], transport-of-intensity equation[14], ptycographic scanning methods[10,15,16] and iterative algorithms[17–20], most of them have to sacrifice the generality of a holographic system by introducing specific assumptions about a sample or an incident beam.

Ideal holography would be a method that directly captures the incident field rather than intensity without any additional constraints (Fig. 1c). One obvious solution is to enhance the bandwidth of a camera above the frequency of light, which is, however, still impractical because the optical frequency is several orders of magnitude faster than the current state-of-the-art electronic techniques.

Here we propose the ideal holographic image sensor using a regular image sensor and a disordered layer and experimentally demonstrate direct holographic measurements of three-dimensional optical fields. To achieve this, we propose a new method called the speckle-correlation scattering matrix (SSM) approach.

## Results

**Hologram retrieval from an intensity speckle pattern.** The basic principle of our method is based on the linear relationship between incident light and scattered light from the diffusive layer, and the randomly distributed (or complex Gaussian distributed) property of the diffused light. The light scattered from a disordered medium is linearly related to the incident light. This linear relationship between incident and scattered light fields is described using a scattering matrix[21]. Previously, measuring the scattering matrix has enabled the imaging or delivering of the designated optical field through a disordered layer[22–25]. However, since the scattering matrix is written by field-field correlation, previous studies had to include additional holographic methods to measure the field. We overcame this limitation by using the random nature of diffused light inspired by the Siegert relation of temporally random (chaotic) light[26] and achieved reference-free holographic imaging using a commercial diffuser as a holographic imaging system (Fig. 1d).

For simplicity without losing generality, we focus on transmission geometry, providing an explanation using a transmission matrix (TM), which is a subset of a scattering matrix. For $N$ orthonormal fields $k_1, k_2, \ldots, k_N$ as the preset input basis, the $p$-th column of TM, $t_p$, is the corresponding diffused field for the $p$-th input field $k_p$ (Fig. 2c–e). Then, for a given arbitrary incident field, $x = \sum_{p=1}^{N} \alpha_p k_p$, the corresponding diffused field can be expressed as $y = \sum_{p=1}^{N} \alpha_p t_p$, where $\alpha_1$, $\alpha_2$, $\ldots$, $\alpha_N$ are complex-valued coefficients that contain the incident field information, and the camera captures the diffused intensity, $y^\star y$ (Fig. 2g). Note that $y^\star y$ loses the wavefront information during the capturing process. Because TM describes the field-field correlation between the input and output light, the field information of diffused light $y$ is required in order to retrieve an incident field $x$ using the TM. Therefore, additional

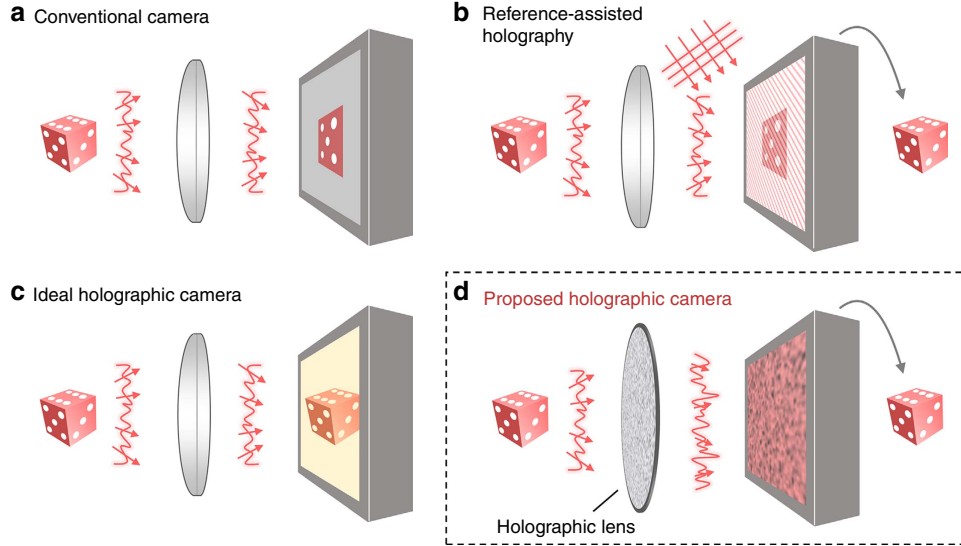

**Figure 1 | Holographic lens as an alternative approach for an ideal holographic image sensor.** (**a**) Regular image sensor loses the wavefront information that contains volumetric perception. (**b**) Reference-assisted holography can deduce the field information via interference with the known reference field. (**c**) An ideal holographic image sensor is identical to the regular image sensor, but records the field information. (**d**) Proposed idea as an alternative approach for an ideal holographic image sensor, utilizing a commercial diffuser as a holographic lens.

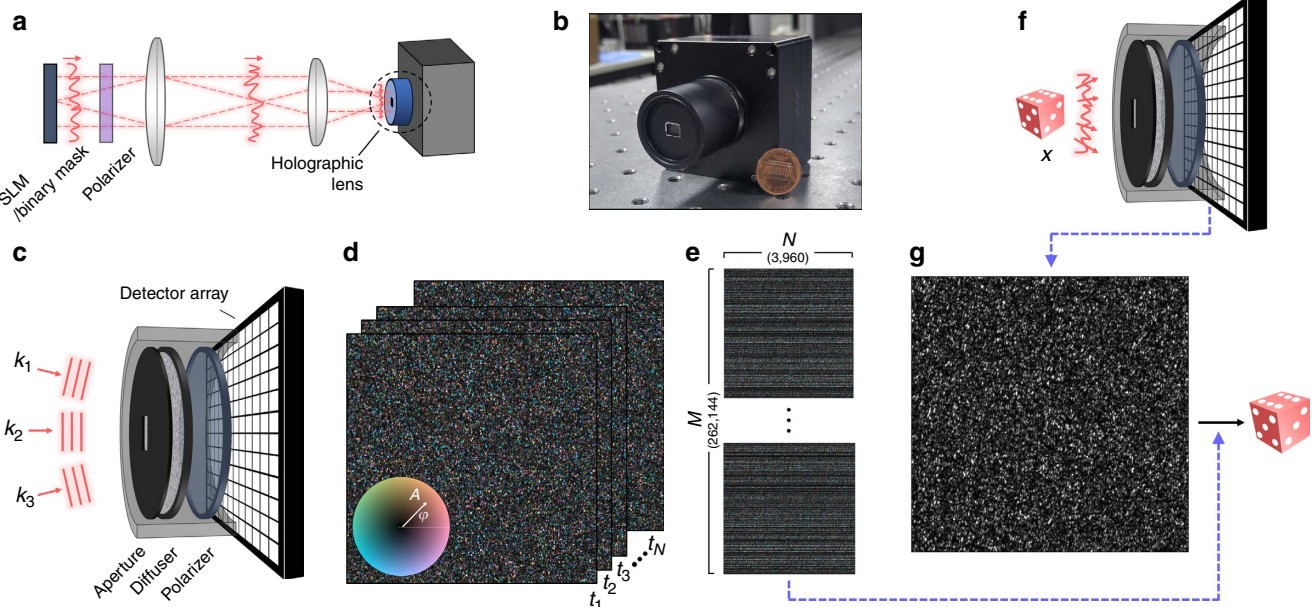

**Figure 2 | Experimental set-up and calibrated transmission matrix. (a)** Schematic of the field measuring optical set-up and the detailed composition and pictures of the proposed holographic image sensor. **(b)** Photogram of the holographic imaging sensor. **(c)** Schematic of the transmission matrix measurement. **(d)** Transmission matrix, which is the collection of scrambled fields for all input modes. In the colour circle, the $A$ and $\varphi$ symbols denote the normalized amplitude and phase, respectively. **(e)** Visualization of the calibrated transmission matrix. **(f)** An unknown incident field $x$ is impinged onto the holographic imaging sensor. **(g)** Incident field retrieval from the field intensity speckle snapshot, $y^*y$.

holographic methods have to be employed in conventional studies using TMs[22–25].

Here, exploiting the random property of diffused fields, we find that calibrated TM, $t_1, t_2, \ldots, t_N$, and the intensity snapshot, $y^*y$, are sufficient for the incident field retrieval. We propose a speckle-correlation scattering matrix $\mathbf{Z}$,

$$\mathbf{Z}_{pq} = \frac{1}{\Sigma_p \Sigma_q} \left[ \left\langle t_p^* t_q y^* y \right\rangle_r - \left\langle t_p^* t_q \right\rangle_r \left\langle y^* y \right\rangle_r \right], \quad (1)$$

where $\langle \cdot \rangle_r$ indicates an space average, and $\Sigma_p = \langle |t_p|^2 \rangle_r$ is a normalization constant. The random property of diffused light field enables an intriguing mathematical relation between fourth-order and second-order moments[27–29],

$$\langle E_1 E_2 E_3 E_4 \rangle = \langle E_1 E_2 \rangle \langle E_3 E_4 \rangle + \langle E_1 E_3 \rangle \langle E_2 E_4 \rangle + \langle E_1 E_4 \rangle \langle E_2 E_3 \rangle, \quad (2)$$

where $E_1$, $E_2$, $E_3$ and $E_4$ are any randomly diffused light fields, and $\langle \cdot \rangle$ indicates the ensemble average. Applying equation (2) to the first term of equation (1), we obtain

$$\mathbf{Z}_{pq} = \alpha_p \alpha_q^* + \frac{1}{\Sigma_p \Sigma_q} \left\langle t_p^* y^* \right\rangle_r \left\langle t_q y \right\rangle_r, \quad (3)$$

where $\alpha_p = \frac{1}{\Sigma_p} \left\langle t_p^* y \right\rangle_r$ from the general orthogonality between uncorrelated diffused fields, $\frac{1}{\sqrt{\Sigma_p \Sigma_q}} \left\langle t_p^* t_q \right\rangle_r = \delta_{pq}$. Although the assumption of orthogonality has generally been accepted for transport through a diffusive layer[23,30,31], correlation exists for highly scattering media[22,32–35], according to the random matrix theory[36,37]. Nonetheless, we note that this correlation can be ignored for a diffusive layer, which is used in this study. If it exists, these correlated elements can be numerically filtered out once a TM is calibrated and its eigenbasis is analysed (See Supplementary Note 1). Because the second term in equation (3) comprises the spatial average of other random fields $t_p^* y^*$ and $t_q y$, it vanishes as the ratio ($\gamma$) between the number

of optical sampling modes ($M$) and the number of preset input modes ($N$) increases (see Supplementary Note 2). Therefore, for a sufficiently large $\gamma$, the second term of equation (2) vanishes, and the rank of $\mathbf{Z}$ matrix becomes unity; $\mathbf{Z}_{pq} = \alpha_p \alpha_q^*$, whose sole eigenvector is the incident field.

**Experimental verification.** To demonstrate the proposed idea, we experimentally construct the set-up (Fig. 2a,b). In addition to a diffuser, we insert an aperture to block ambient light, and a polarizer to maximize the speckle visibility. To avoid duplicated optical information, we set the speckle grain size at the camera plane to be comparable to a single camera pixel by adjusting the distance between the diffuser and the camera. Using a He-Ne laser (wavelength, 633 nm), the TM is measured by a spatial light modulator (SLM) (see Methods).

Because the retrieved hologram is mapped onto the preset input basis, the number of input modes $N$ determine the field of view (FOV) and the diffraction-limited spot size (that is, the resolution) of the holographic image sensor. The number of optical sampling modes $M$ relates to the intrinsic signal-to-noise ratio (SNR) originating from the second term of equation (3). In our system, we used $66 \times 60$ ($N = 3,960$) equally spaced spatial frequency modes as an input basis. This corresponds to the FOV of $5.28 \times 4.00$ mm and a diffraction-limited spot size of $80 \times 67\,\mu$m. Additionally, we set $512 \times 512$ central camera pixels ($M = 262,144$) as the output modes, which corresponds to $\gamma = 66.2$.

First, we test the feasibility of our idea by imaging incident fields, which are modulated by a SLM (wavefront modulation) or a binary mask (amplitude modulation). We use a 4-$f$ telescopic imaging system with a demagnification factor of 3 to conjugate the focal position of the holographic image sensor to the position of the SLM or the binary mask, and also to increase the effective FOV. The $\mathbf{Z}$ matrix is calculated by substituting a calibrated TM and a measured $y^*y$ (Fig. 3a) into equation (1).

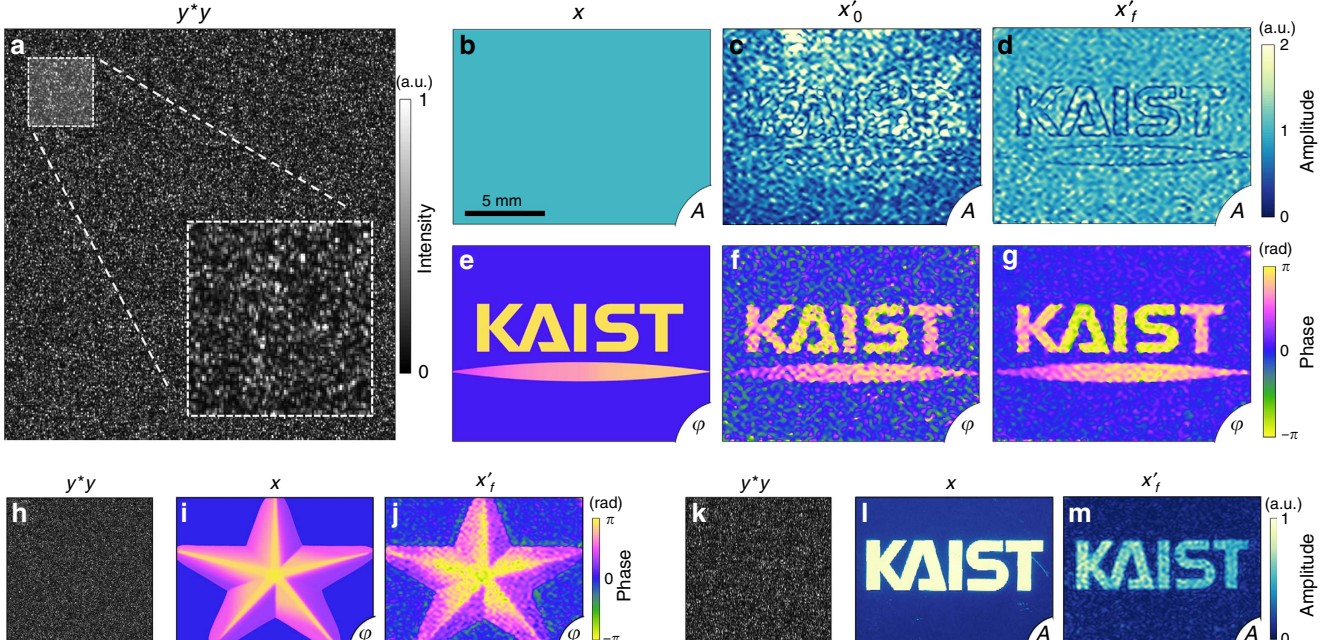

**Figure 3 | Experimental demonstrations for known incident fields.** (**a**–**g**) For a given incident field ($x$; (**b**,**e**)), a raw intensity speckle single shot ($y^*y$; (**a**)), the retrieved hologram using the **Z** matrix ($x'_0$; (**c**,**f**)) and the final hologram after the iteration ($x'_f$; (**d**,**g**)). (**h**–**m**) Additional demonstrations of the proposed method with wavefront-modulated (**h**–**j**), and amplitude-modulated (**k**–**m**) incident fields. Note that the amplitude and wavefront part of the holograms are labelled with the respective symbols $A$ and $\varphi$ on the bottom right corner of the figures.

The amplitude and wavefront results of the retrieved hologram $(x'_0)$ are shown in Fig. 3c,f, respectively. Despite the presence of noise mainly originating from the second term of equation (3), the expected field (Fig. 3b,e) is successfully reconstructed.

To suppress speckle noise, we additionally present a modified Gerchberg-Saxton algorithm (see Methods, Supplementary Movie 1). We use a measured TM as an operator, instead of Fourier transformation, and adopt $|y|$ as a constraint. Starting from $x'_0$, the iteration process stops when it converges into $x'_f$. Note that the proposed iteration algorithm does not require any additional assumptions, free variables or regularizations. The iteration result is shown in Fig. 3d,g. Since only a limited portion of the Fourier modes is recorded in the TM, the effects of low-pass filtering are observed in the retrieved amplitude (Fig. 3d). In addition, several wavefront- and amplitude-modulated fields are tested (Fig. 3h–m).

**The noise tolerance**. To analyse the noise tolerance of the proposed method, we performed numerical simulations (Fig. 4). We added Gaussian (white) noise to both TMs, and measured the speckle intensity maps by adjusting $\gamma = M/N$ (or the intrinsic noise level) and the SNR of Gaussian noise. In order to systematically investigate the effects of noise, the correlation between the incident field ($x$, ground truth) and the retrieved field ($x'_0$ or $x'_f$) is quantified for both the SSM and the SSM with the iteration method. As quantitatively analysed in Fig. 4d,e, the validity of the present method is satisfied when the scattering correlation matrix is appropriately sampled and measured; that is, $\gamma$ is set to be large. Importantly, the present method provides the successful retrieval of holographic images even in extremely noisy conditions (for example, SNR = 1) when $\gamma$ is large enough. Moreover, for a small $\gamma$ (for example, $\gamma = 4$), the modified GS algorithm significantly enhances the quality of the retrieved hologram when the SNR is sufficiently large (for example,

SNR > 10). For all numerical simulations, we used a fixed $N = 1,024$, while $M$ varied depending on $\gamma$. The detailed procedures for the numerical implementations and the used MATLAB code can be found in Supplementary Methods.

**The hologram of real objects**. To demonstrate the versatility of our method as a stand-alone 3D holographic image sensor, we prepare a real target consisting of two diffusive dices with a size of $5 \times 5 \times 5$ mm, which are separated by 40 cm from each other (Fig. 5a). We impinge a laser beam onto the target for illumination and measure the reflected light field (Fig. 5b–d). To confirm the quality of the measured hologram, we perform numerical refocusing from $-35$ to $+35$ cm, as shown in Fig. 5e–k. As shown, the results exhibit the expected silhouette of targets at each refocusing position. Note that the speckled intensity and randomized phase (Fig. 5c,d) are a result of the diffusive reflection of an object[38]. For clear visualization, the speckle is suppressed by measuring 25 more holograms with slightly tilted laser illumination. For a given refocusing position, all 25 retrieved holograms are numerically propagated to the position, and their amplitude parts are compounded. The compounding results shown in Fig. 5l–r show clearer images of the target for every numerically refocused position. These results prove the capability of our holographic method, even for the complex form of the incident field. Once the TM of the system is calibrated, the system can be functional as long as the physical structures are not altered. We observe no significant degradation in the quality of holographic imaging, even 1 month after the calibration without any additional temperature or humidity controls.

In conclusion, we theoretically propose and experimentally demonstrate a reference-free holographic imaging sensor. The present method enables the direct retrieval of the phase without using reference-beam-assisted interferometry. Exploiting the random property, light-field information can be directly

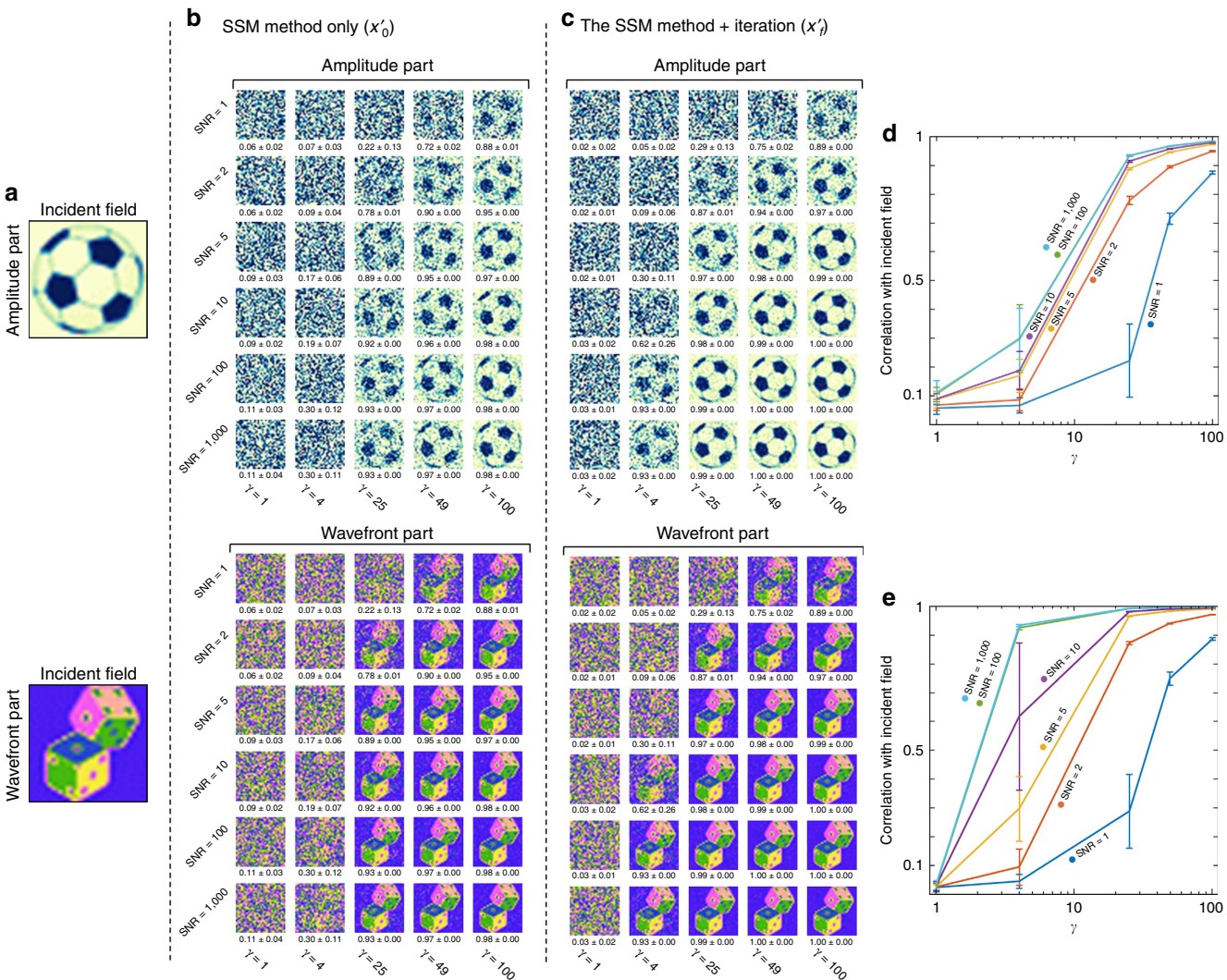

**Figure 4 | The effects of noises on the holographic measurements.** (**a**) The amplitude (top) and the wavefront (bottom) part of the tested incident field. (**b,c**) The retrieved hologram using the speckle-correlation scattering matrix (SSM) method only (**b**) and the SSM method with the iteration algorithm; the mean and s.d. of the correlations with the incident field are denoted below each result (**c**). (**d,e**) The correlation between the incident field and the retrieved holograms as a function of $\gamma$ and the white noise signal-to-noise ratio for the SSM method only (**d**), and the SSM method with the iteration algorithm (**e**). Error bars correspond to the s.d. of 20 correlation results for each data point.

obtained from the measurement of a diffused intensity image. The principle is that light transport through an optical diffusor is described with a random matrix, and the intensity speckle patterns from two different input wavefronts are effectively not correlated with each other. To demonstrate the proposed idea, we present a compact and reference-free holographic image sensor, with which direct measurements of various light-field images are shown.

We should emphasize that the present approach is fundamentally different from previously reported methods based on speckle correlation or ghost imaging. For example, conventional speckle correlation methods such as ghost imaging[39–42], single-pixel imaging[43–46], memory effect-based imaging[31,32] and Siegert-relation-based imaging[47,48] have been able to retrieve only amplitude images, because their speckle correlation is based on intensity correlation, $\langle t^*ty^*y \rangle_r$; therefore, the wavefront information is inherently lost. In addition, unlike existing TM-based methods[22–25], the current work does not require the use of a reference beam, which significantly expands the applicability of our proposed method. Our method also does not require *any* type of prior knowledge of incident light, including reference field, which enables direct hologram capture as photography does. From a technical point of view, the present method can be extended to image polarization- and wavelength-dependent imaging, as well as subwavelengths near-field imaging, by exploiting the large degree of freedom that multiple light scattering offers[49–52]. Although the present method requires a precise and time-consuming calibration step of a turbid layer prior to use, faster techniques for measuring TMs will further alleviate this technical issue.

Therefore, we expect that our method could be a solution for applications with which conventional holographic approaches have had difficulties. Furthermore, the assumption-free and reference-free capability of the present method would be advantageous and could also be expanded to direct holographic measurement in an X-ray regime. The compactness and single-shot property would be beneficial for more practical applications such as a handheld holographic camera using smartphone optics[53] or quantitative-phase imaging[8] for point-of-care biomedical diagnostic applications.

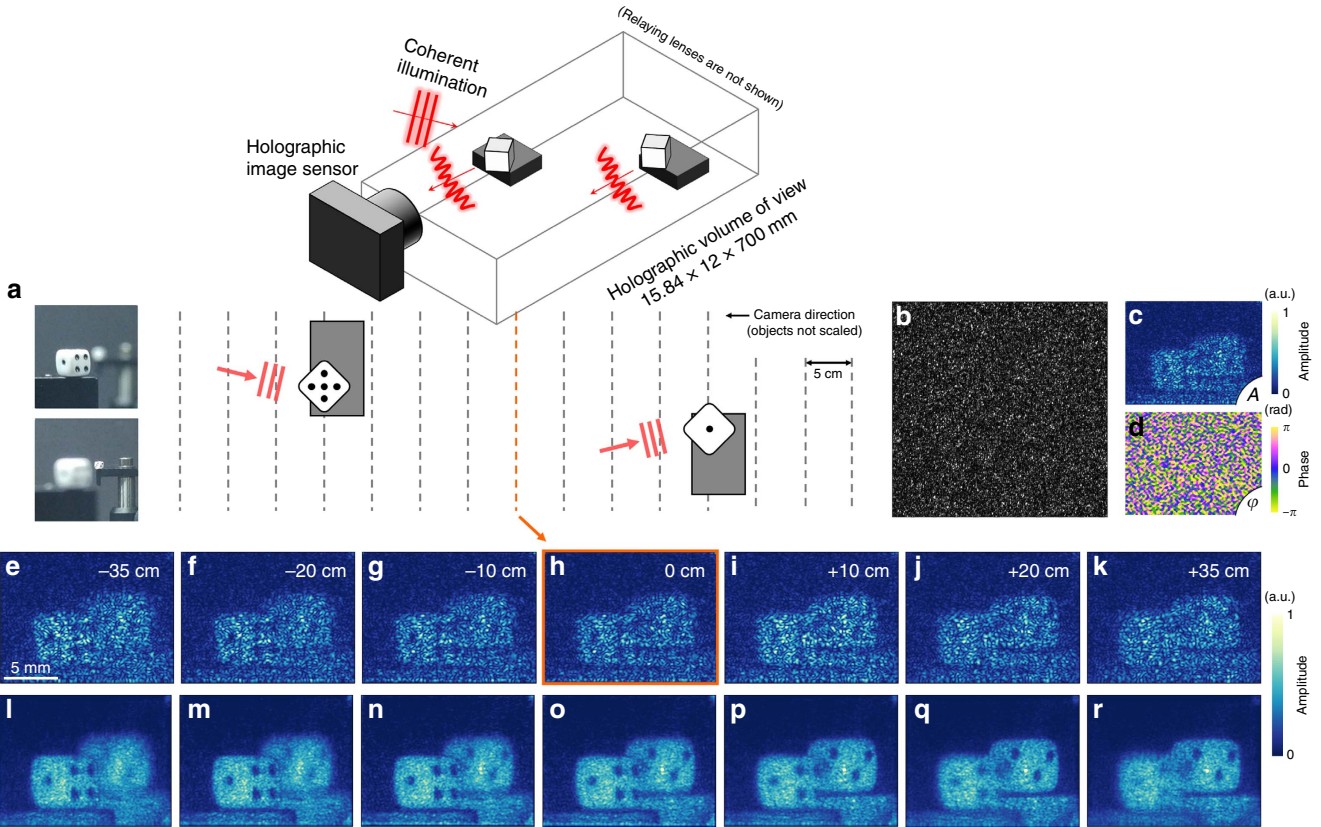

**Figure 5 | Diffusive field retrieval.** (**a**) Schematic of the diffusive target; the pictures were taken by a photographic camera focusing on each dice. (**b–d**) A speckle intensity image (**b**) and the amplitude (**c**) and phase image (**d**) of the retrieved hologram. (**e–k**) Numerically refocused results from a single retrieved hologram. (**l–r**) Angular compounded results at each refocused position.

## Methods

**Holographic image sensor.** We use a lens tube (SM1L10, Thorlabs Inc.) as the body of the holographic lens. From the front, a rectangular-shape aperture (4 × 5.3 mm, custom-made), an optical diffuser (ED1-C20, Thorlabs Inc.) and a polarizer (LPVISE100-A, Thorlabs Inc.) are sequentially stacked inside the tube. The direction of the polarizer is set to be parallel to the polarization of the incident laser. We directly assemble the lens with a camera (4,242 × 2,830 pixels; 3.1 μm pitch; MD120MU-SY, XIMEA GmbH) using an adapter (SM1A9, Thorlabs Inc.). The distance between the diffuser and the detector array is estimated to be 43 mm, including the additional space inside of the camera.

**TM calibration.** Overall, we employed an improved version of the TM calibration algorithm suggested in ref. 54. As shown in Supplementary Fig. 1, we construct a Michelson interferometer using a He-Ne laser (HNL050L, Thorlabs Inc.) with a SLM in the sample arm. After fixing the interferometer, we demagnify the beam by a factor of 3 × using a 4-$f$ lens system in order to fit the clear aperture of the SLM (792 × 600 pixels; 20 μm pitch; X10468-01, Hamamatsu photonics K.K.) into the size of the aperture. We use 66 × 60 rectangular ($N = 3,960$) low-pass Fourier modes as an input basis, which corresponds to a FOV of 5.28 × 4.00 mm, and a resolution of 80 × 67 μm; the output modes are 512 × 512 central camera pixels ($M = 262,144$). For each $p$-th input mode, we take three interferograms $I_{0,p}$, $I_{1,p}$ and $I_{2,p}$, adding respective global phase delays of 0, $\frac{2\pi}{3}$ and $\frac{4\pi}{3}$. Utilizing the phase retrieval algorithm, we retrieve the output field $t_i$ from the phase-shifted interferograms $R^* t_p = \frac{1}{3}\left[I_{0,p} - \frac{1}{2}I_{1,p}\left(1 + \frac{1+\sqrt{3}}{2}i\right) - \frac{1}{2}I_{2,p}\left(1 - \frac{1+\sqrt{3}}{2}i\right)\right]$, where $R$ indicates the reference field, and $i = \sqrt{-1}$. Since the reference field also passes through the holographic unit, $R$ also fluctuates along the space, which causes severe distortion, which is the result of $Z$ matrix in equation (2). To compensate for the effect of the $R$, we take one additional image while the SLM arm is blocked, which is $|R|^2$. Note that the phase part of $R$ is unnecessary because it is automatically compensated for during the $Z$ matrix calculation. As a result, we take a total of $3N + 1$ (11,881) images to calibrate the TM. Our TM calibration takes 20 min. Please refer Supplementary Methods for the detailed flow chart and full MATLAB codes.

**Modified Gerchberg-Saxton (GS) iteration algorithm.** The Fourier transform operator in the original GS iteration algorithm is replaced with a measured TM. Since we set the constraint $|y|$ and starting point $x'_0$, the rest of the procedure is identical to the field retrieval sequence of a single-constraint situation of an

error-reduction iteration algorithm[18]. We stop the iteration at the $f$-th loop satisfying the condition that the correlation between $x'_{f-1}$ and $x'_f$ is above 0.999998. The iteration number widely ranged from 34 to 130 proportional to the noise signal (see Supplementary Fig. 2). The mean iteration numbers of the results in Figs 2 and 3 are 40 and 74, respectively. It took 1.75 s for a single iteration step with our computing power (3.50 GHz, intel core i5-4690 CPU; 32.0 GB RAM) using the MATLAB software. Please refer Supplementary Methods for the detailed flow chart and full MATLAB codes.

**Data availability.** The data that support the findings of this study are available from the corresponding author upon reasonable request.

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

## Acknowledgements

This work was supported by KAIST, and the National Research Foundation of Korea (2015R1A3A2066550, 2014K1A3A1A09063027, 2014M3C1A3052567) and Innopolis foundation (A2015DD126).

## Author contributions

K.L. developed the theory, performed the experiments, and analysed the data. Y.P. conceived and supervised the project. K.L. and Y.P. wrote the manuscript.

## Additional information

**Competing financial interests:** K.L. and Y.P. are inventors on a patent describing a method for direct holography (Republic of Korea patent application number 10-2016-0043274).

**Publisher's note**: 

