## [Peer Review File · Nature Communications]

Reviewers' Comments:

Reviewer #1 (Remarks to the Author):

This paper is an interesting variation on well-established methods of correlation optics and cross-correlation imaging. These are based the use of random light to illuminate the object, and measurement of the cross correlation of the image intensity with a copy of the intensity of the random illumination. Under the Gaussian assumption, intensity correlation is related to the field correlation by an equation similar to Eq. (2), which leads to the well-known Siegert relation. This literature includes ghost imaging (both classical and quantum), correlation holography (see e.g., Takeda, Optics Express 13, 96, 2005)), and dual photography (see, e.g., Sun et al, Science, 340, 846, 2013). Unfortunately, none of this extensive literature is cited!

In this paper, the random illumination is replaced by modulation of the field scattered from the object with elements of a random scattering matrix, and instead of measuring the cross correlation of the resultant intensity (called yy^* in this paper) with the intensity of the random field (tt^*), here the cross correlation is between yy^* and field products of orthogonal pairs of the random fields (called tp^*tq in this paper). Also, as I understand it, the novelty in this paper is in the fact that the noise field is assumed to be known (magnitude and phase) since the scattering matrix is assumed to be known. The author does not tell us this explicitly, however.

I do not recommend publication of this paper in Nature Communication for the following reasons.

1) It ignores the extensive prior literature on correlation optics (although it cites old work on conventional holography and the phase problem). 2) It fails to identify the novelty of the contribution and does not provide a comparison with existing methods. 3) The paper is very difficult to read since the mathematical description of the method is not properly related to the physical setup.

Reviewer #2 (Remarks to the Author):

A) Summary of the key results

In the manuscript entitled "Exploiting speckle-correlation scattering matrix: an approach to a compact reference-free holographic image sensor", the authors demonstrate a reference-free holographic image sensor by using a speckle-correlation scattering matrix approach. The light field information passing through a thin disordered layer is recorded and retrieved from a single-shot recording of speckle intensity patterns.

B) Originality and interest: if not novel, please give references

While the calibration algorithm used is just an improved version of that reported in Ref. 1, its application for the holographic measurement of 3D wavefields is novel.

C) Data & methodology: validity of approach, quality of data, quality of presentation

The theory is OK and the presented results clearly show that the proposed method is able to reconstruct 3D optical fields. The results are well presented.

D) Appropriate use of statistics and treatment of uncertainties

It would be nice if the authors could provide some information on the Signal/Noise ratio of their reconstructions.

E) Conclusions: robustness, validity, reliability

At lines 132-133 and 147-152 the authors say that their method "will find immediate applications in various optical applications".

It seems to me that this claim is exaggerated. It is true that their approach has some advantages but in my opinion they should objectively describe the advantages and disadvantages of their method compared with digital holography and other phase retrieval techniques that do not require a reference.

It is true that they do not require a reference but a calibration process is necessary.

F) Suggested improvements: experiments, data for possible revision

From the results shown in Fig. 4 it looks like that the system has very low numerical aperture (long depth of focus). Could be possible to have higher numerical apertures?

The caption in Fig. 4 is wrong, 4. b,c and d are not "numerically focused results". It seems that 4e-k are "numerically focused results" and 4l-r are clearer images obtained by measuring 25 or more holograms with slightly tilted illumination.

They say that the field information is recorded in a single shot but it seems that for obtaining good quality results more holograms with tilted illumination are necessary.

Is their method suitable for the measurement of large objects?

G) References: appropriate credit to previous work?

Appropriate credit is given to previous work.

H) Clarity and context: lucidity of abstract/summary, appropriateness of abstract, introduction and conclusion

The paper is clearly written but the claims are exaggerated.

Reviewer #3 (Remarks to the Author):

One of the main advantages that doing experiments in the microwave regime has over the same kind of experiments in the optical regime, is that modern electronic is fast enough to measure directly the electric field of microwaves, but it can not do the same for visible light. And since it is extremely unlikely that electronics will ever become fast enough to do it, people working in the optical regime are left with a huge problem: intensities are the only quantity that can be measured directly. Measuring the phase of a visible wavefront can be accomplished using one of the many possible flavours of interferometry, but most of them require a reference beam (but not all, see e.g. Drémeau et al. Optics Express 23, 11898, 2015), which complicate setup.

In this manuscript the authors exploit the fact that an optical diffuser can completely scramble a wavefront to recover the missing phase information. The basic idea is that the transmission matrix of an optical diffuser is effectively a random matrix, making it exceedingly unlikely that two different input wavefronts will give rise to the same intensity speckle pattern. The authors formalize this concept and prove experimentally that recovering the phase via this method is indeed possible. I am a bit puzzled at why they claim that they are using "speckle intensity correlations", while in fact the whole construction rely on the assumption that the elements of the

transmission matrix are completely uncorrelated and thus $\langle t^*_p t_q \rangle$ averages to a delta, but the general idea is solid and the experimental proof convincing.

What in my opinion is missing is a bit more discussion about the method implementation. As stated above the whole method rely on the fact that the elements of t are uncorrelated, but very often "measured" transmission matrices do not have uncorrelated elements at all! (see ref 21 and 22) Of course these spurious correlations can be minimized, but they are never zero, and I think the manuscript would hugely benefit of a proper discussion of the impact of these correlations of the final output. At least in the supplemental informations if the authors do not want to go into details in the main body of the manuscript. Similarly it would be useful and instructive to know how the results are expected to change if one increases either N or M . Is it worthwhile to maximise those numbers, or there is a trade off? The authors also never really discuss the fact that it is effectively impossible to measure the whole transmission matrix, and one always has to be happy with a subset of it. How big is the fraction of the matrix element the authors can measure? Does it make sense to push for a large fraction? Does the fact that correlations will appear if one measure a large fraction of the transmission matrix create a limit on the possible resolution of this method? Furthermore, what about stability? What happens is the diffuser, once characterized, is not perfectly aligned? How much tolerance does this method allows for? All these questions deserve some thought and discussion, especially because the authors are writing a paper on a novel technique that they hope will be used by others.

Finally I would like to note that the presented technique is very computational in nature. Contrary to what fig. 1d suggests, this technique does not create a holographic image on a detector, but the only measured quantity is an apparent mess which can then be used to computationally recover the desired amplitude and phase. This is absolutely not a problem, but it would be nice to see it acknowledged in the manuscript.

In conclusion this manuscript presents an intriguing and novel technique to measure the phase of an unknown wavefront, but leaves a large number of unanswered questions, that must be addressed before anyone can seriously think about implementing this method as a research tool.

Reviewer #1 (Remarks to the Author):

This paper is an interesting variation on well-established methods of correlation optics and cross-correlation imaging. These are based the use of random light to illuminate the object, and measurement of the cross-correlation of the image intensity with a copy of the intensity of the random illumination. Under the Gaussian assumption, intensity correlation is related to the field correlation by an equation similar to Eq. (2), which leads to the well-known Siegert relation. This literature includes ghost imaging (both classical and quantum), correlation holography (see e.g., Takeda, Optics Express 13, 96, 2005), and dual photography (see, e.g., Sun et al, Science, 340, 846, 2013). Unfortunately, none of this extensive literature is cited!

We thank the reviewer for careful reading and the suggestion. The reviewer was correct that the Siegert relation can be derived from Eq. (2). However, the key novelty of this work is that the phase information of the object can be directly retrieved without a reference beam, by exploiting multiple scattering in the turbid layer with the presented new approach – the speckle-correlation scattering matrix. We have included the suggested references in the revised manuscript, and also discussed the novelty of this work in the revised manuscript as well as the detailed responses below.

In this paper, the random illumination is replaced by modulation of the field scattered from the object with elements of a random scattering matrix, and instead of measuring the cross correlation of the resultant intensity (called yy^* in this paper) with the intensity of the random field (tt^*), here the cross correlation is between yy^* and field products of orthogonal pairs of the random fields (called tp^*tq in this paper).

As the reviewer addressed, our work has common intersections with other correlation-based imaging techniques such as ghost imaging, single-pixel imaging, and the Siegert-relation based imaging. However, we would like to emphasize that our method is the first suggestion and realization of single-shot reference-less holography that reconstructs not only amplitude but also *wavefront* information of incident light.

Importantly, due to the well-known ‘*phase problem*’, existing correlation-based imaging methods have been forced only to consider the *intensity* correlation (that is, $\langle t^*ty^* \rangle$ according to the reviewer’s comment).

1. Ghost imaging (Refs. 39-42 in the revised manuscript) and single-pixel imaging (Refs. 43-46) reconstruct only the *intensity* image of a target using the intensity correlation with patterned illuminations and a patterned aperture, respectively. The *phase* information cannot be retrieved using conventional ghost imaging methods. For this reason, the conjugation of 2-D intensity patterned illumination or aperture with an object is crucial for these techniques to recover a well-focused *intensity* image, since the intensity pattern varies as the light propagates. For example, in Ref. 42, the authors equalized the distance between a beam splitter and a camera with the distance between the beam splitter and the target object (“...When the CCD was moved away from $d_B = d_A$, the images were blurred...”).
2. As the reviewer addressed, the Siegert relation can be derived from Eq. (2), by selecting $E_1 = \psi_1^*$, $E_2 = \psi_1$, $E_3 = \psi_2^*$, and $E_4 = \psi_2$ (Ref. 29),

$$\begin{aligned}\langle \psi_1^* \psi_1 \psi_2^* \psi_2 \rangle &= \langle \psi_1^* \psi_1 \rangle \langle \psi_2^* \psi_2 \rangle + \langle \psi_1^* \psi_2^* \rangle \langle \psi_1 \psi_2 \rangle + \langle \psi_1^* \psi_2 \rangle \langle \psi_1 \psi_2^* \rangle \\ &= \langle |\psi_1|^2 \rangle \langle |\psi_2|^2 \rangle + |\langle \psi_1^* \psi_2 \rangle|^2.\end{aligned}$$

Please note that wavefront information of both ψ_1 and ψ_2 are exactly compensated by its conjugated term in the left-hand side of the equation; the right-hand side of the equation also has no imaginary values. Thereby, the Siegert-relation-based imaging (Refs. 47,48) only has been able to retrieve the intensity information of the incident field. For example, in Ref. 48, the authors should have translated the camera along the z -direction to obtain a photography of different depth (“... *With the help of a stepper motor stage, the camera can be moved along the z direction to scan the speckle along z direction...*”).

Also, as I understand it, the novelty in this paper is in the fact that the noise field is assumed to be known (magnitude and phase) since the scattering matrix is assumed to be known. The author does not tell us this explicitly, however.

The novelty of this work is in providing a method that enables the holographic measurement of an incident field without additional reference beams or assumptions. This method exploits (1) the linear relationship between the incident and scattered light from complex media and (2) the random property of diffused light. By measuring a speckle intensity image that is generated when a light field of interest passes through a calibrated turbid layer, the field information of an arbitrary optical field can be directly reconstructed. Importantly, we would like to note that the scattering matrix was *not* “assumed to be known”, indeed it is precisely measured. We admit this misunderstanding was partially caused by the insufficient information about the methods in the previous manuscript. Thus, in the revised manuscript and supplementary information, we have included the detailed algorithm, procedures, and MATLAB code to explicitly explain the principle.

I do not recommend publication of this paper in Nature Communication for the following reasons. 1) It ignores the extensive prior literature on correlation optics (although it cites old work on conventional holography and the phase problem). 2) It fails to identify the novelty of the contribution and does not provide a comparison with existing methods. 3) The paper is very difficult to read since the mathematical description of the method is not properly related to the physical setup.

We thank the reveiwer for the valuable comments. We fully agree that our previous manuscript had insufficient information regarding the raised concerns. In the revised manuscript, we have addressed the comments as follows.

1. The discussion part now addresses detailed comparisons with existing correlation imaging techniques with suggested references (Lines 164-169 and Refs. 39-48).
2. We elaborate on the present method to discuss its novelty, including comparisons with the existing methods.
3. We have included additional formation about the detailed realizations of the proposed method (Supplementary Note 3), and the MATLAB code for numerical simulations (Supplementary Note 4).

We believe that we have addressed all of the referee’s criticisms and these modifications have strengthened the manuscript.

Reviewer #2 (Remarks to the Author):

A) Summary of the key results

In the manuscript entitled "Exploiting speckle-correlation scattering matrix: an approach to a compact reference-free holographic image sensor", the authors demonstrate a reference-free holographic image sensor by using a speckle-correlation scattering matrix approach. The light field information passing through a thin disordered layer is recorded and retrieved from a single-shot recording of speckle intensity patterns.

We appreciate the brief summary of our work.

B) Originality and interest: if not novel, please give references

While the calibration algorithm used is just an improved version of that reported in Ref. 1, its application for the holographic measurement of 3D wavefields is novel.

We thank the reviewer for appreciating the novelty of our work. In addition, we note that the key originality of this method is that it can measure 3D wavefields without introducing a reference beam by exploiting the chaotic nature of diffuse light.

C) Data & methodology: validity of approach, quality of data, quality of presentation

The theory is OK and the presented results clearly show that the proposed method is able to reconstruct 3D optical fields. The results are well presented.

We appreciate the kind compliments on our work.

D) Appropriate use of statistics and treatment of uncertainties

It would be nice if the authors could provide some information on the Signal/Noise ratio of their reconstructions.

We thank the reviewer for the helpful comment. As the reviewer suggested, we performed numerical simulations to test the tolerance of our method in practical (noisy) situations. (lines 125-138, and Fig. 4 in the revised manuscript)

To analyze the noise tolerance of the proposed method, we performed numerical simulations. We added Gaussian (white) noise to both the transmission matrix (TM) and the measured intensity speckle. By adjusting $\gamma = M/N$ (or the intrinsic noise level) and the signal-to-noise ratio (SNR), we systematically tested the effects of noise on the present method. The correlation between the incident field (x , ground truth) and the retrieved field (x'_0 or x'_f) are quantified. For all numerical simulations, we used a fixed $N = 1024$, while M varies depending on γ . Please find the full MATLAB code used in the numerical simulations in Supplementary Note 4.

I. Retrieved field using SSM method only (x'_0 ; without the iteration process)

II. Retrieved field using the iteration process (x'_f)

III. Correlation between the incident field and the retrieved field as a function of M to N ratio (γ) and SNR.

The results clearly show the validity of the present method when the scattering correlation matrix is appropriately sampled and measured; i.e. γ is set to be large. Importantly, the present method provides successful retrieval of holographic images even in extremely noisy situations (e.g. SNR=1) when γ is large enough. Also, even with a small γ (e.g. $\gamma = 4$), the modified GS algorithm significantly enhances the quality of hologram retrieval when the SNR is sufficiently large (e.g. SNR > 10).

IV. Mean number of iterations vs. M to N ratio (γ) and SNR.

We also found the total number of iterations decreases as γ increases. However, because a large γ requires a large transmission matrix for a given N , the computation time for each iteration would proportionally increase.

E) Conclusions: robustness, validity, reliability

At lines 132-133 and 147-152 the authors say that their method “will find immediate applications in various optical applications”. It seems to me that this claim is exaggerated. It is true that their approach has some advantages but in my opinion they should objectively describe the advantages and disadvantages of their method compared with digital holography and other phase retrieval techniques that do not require a reference. It is true that they do not require a reference but a calibration process is

necessary.

We thank the reviewer for the comment, and we are in agreement. As suggested, we toned down the claim regarding the future application and also discussed the limitations of the present method (lines 173-185 in the revised manuscript).

F) Suggested improvements: experiments, data for possible revision

From the results shown in Fig. 4 it looks like that the system has very low numerical aperture (long depth of focus). Could be possible to have higher numerical apertures?

Yes, it is possible to have a higher numerical aperture (NA). Because the retrieved hologram is mapped on the input basis (spatial frequency, k_1, k_2, \dots, k_N), the numerical aperture (NA) and the field-of-view (FOV) of the current method depend on the number of input basis of the calibrated transmission matrix. For example, if the input basis are chosen as a set of equispaced spatial frequencies, simply widening the spacing between basis vectors can increase the NA while the FOV decreases as a trade-off (according to the Nyquist theorem).

Even though increasing the number of input basis (N) will increase both the FOV and the NA, $N = 3,960$ turned out to be an appropriate choice for our first experimental system due to several practical issues such as camera acquisition speed and computation power. Thus, all experimental results are based on a set of 66×60 equispaced spatial frequency basis with a fixed FOV = $5.28 \text{ mm} \times 4.00 \text{ mm}$. This corresponds to very low NA = 0.0040×0.0047 for 633 nm wavelength (or the diffraction-limited spot size of $80 \mu\text{m} \times 67 \mu\text{m}$).

In the revised manuscript, we have provided more detailed descriptions of the NA and the FOV (lines 100-107).

The caption in Fig. 4 is wrong, 4. b,c and d are not “numerically focused results”. It seems that 4e-k are “numerically focused results” and 4l-r are clearer images obtained by measuring 25 or more holograms with slightly tilted illumination.

We appreciate the reviewer pointing out this mistake. The caption is corrected in the revised manuscript.

They say that the field information is recorded in a single shot but it seems that for obtaining good quality results more holograms with tilted illumination are necessary.

The reason we used several holograms with tilted illuminations is that we measured an object with diffusive surfaces with a coherent illumination, not because of noise or inherent limitations of the present method. As is well known in the field of holographic imaging (Ref. 38), the speckle patterns and randomized phase images in Fig. 5 are naturally expected. In general, measuring more holograms with tilted illumination is not necessary for general applications. However, we included the results with more holograms with tilted illumination only for the purpose of clearer visualization. In the revised manuscript, we have clearly expressed this issue.

Is their method suitable for the measurement of large objects?

Yes, the field-of-view (FOV) can be further increased to measure large objects. This is related to the previous comment regarding the NA. Similar to general imaging systems, the relationship of the space-band-product is also applied to this method; the product of the NA and the FOV is a constant, which is determined by the pixel resolution of an imaging system (N in our method). For a fixed N , the FOV can be increased at the expense of the NA. To increase both the FOV and the NA, one should increase N . Alternatively, one can readily increase the FOV using the spatial multiplexing method; i.e. stitch multiple holograms that are measured by translating the holographic image sensor.

G) References: appropriate credit to previous work?

Appropriate credit is given to previous work.

Thank you for the comment.

H) Clarity and context: lucidity of abstract/summary, appropriateness of abstract, introduction and conclusion

The paper is clearly written but the claims are exaggerated.

As responded above, we have toned down the expression.

Reviewer #3 (Remarks to the Author):

One of the main advantages that doing experiments in the microwave regime has over the same kind of experiments in the optical regime, is that modern electronic is fast enough to measure directly the electric field of microwaves, but it can not do the same for visible light. And since it is extremely unlikely that electronics will ever become fast enough to do it, people working in the optical regime are left with a huge problem: intensities are the only quantity that can be measured directly. Measuring the phase of a visible wavefront can be accomplished using one of the many possible flavours of interferometry, but most of them require a reference beam (but not all, see e.g. Drémeau et al. Optics Express 23, 11898, 2015), which complicate setup.

In this manuscript the authors exploit the fact that an optical diffuser can completely scramble a wavefront to recover the missing phase information. The basic idea is that the transmission matrix of an optical diffuser is effectively a random matrix, making it exceedingly unlikely that two different input wavefronts will give rise to the same intensity speckle pattern. The authors formalize this concept and prove experimentally that recovering the phase via this method is indeed possible.

We appreciate the brief introduction and the appreciation of the novelty of our work. Also, we thank the reviewer for suggesting the relevant paper, which has been cited in the revised manuscript.

I am a bit puzzled at why they claim that they are using "speckle intensity correlations", while in fact the whole construction rely on the assumption that the elements of the transmission matrix are completely uncorrelated and thus $\langle t^*_p t_q \rangle$ averages to a delta, but the general idea is solid and the experimental proof convincing.

We appreciate the kind compliments, careful reading, and valuable comments on our work. We agree that the expression "speckle intensity correlation" does not properly describe our work. We revised the manuscript appropriately reflecting the comment (lines 57-60).

What in my opinion is missing is a bit more discussion about the method implementation. As stated above the whole method rely on the fact that the elements of t are uncorrelated, but very often "measured" transmission matrices do not have uncorrelated elements at all! (see ref 21 and 22) Of course these spurious correlations can be minimized, but they are never zero, and I think the manuscript would hugely benefit of a proper discussion of the impact of these correlations of the final output. At least in the supplemental informations if the authors do not want to go into details in the main body of the manuscript.

We thank the reviewer for bringing our attention to the point. As the reviewer correctly pointed out, there exist (very few) correlated elements in the transmission matrix, even though the assumption of non-correlation has generally been accepted for a diffusive layer (revised manuscript, Refs. 23,30,31). As suggested, we have considered the effects of this correlation to the final output. For simplicity without loss of generality, we considered two cases: (1) without measurement noise and (2) with measurement noise.

- (1) In the absence of measurement noise, the exact orthogonality can always be made by transforming the basis of interest. Once a transmission matrix is calibrated, we can freely transform the basis. The unwanted correlation elements can then be filtered out by simply

selecting the eigenvectors (or left-singular vectors) u_1, u_2, \dots, u_N (instead of t_1, t_2, \dots, t_N) as the basis of interest, which ensures the orthogonality, i.e. no correlation between elements. Since the eigenvectors of the transmission through a diffusive layer are also randomly diffused in general, they also satisfy Eq. (2). Thereby, we can reconstruct a hologram by constructing a Z matrix [Eq. (1)]:

$$\mathbf{Z}_{pq} = \langle u_p^* u_q y^* \rangle_r - \langle u_p^* u_q \rangle_r \langle y^* y \rangle_r,$$

where $\sum_{p,q} \langle |u_{p,q}|^2 \rangle_r = 1$ in this case. Even if the transmission matrix is not a full-rank matrix (i.e. $\text{rank} < N$), our method is still valid. However, the effective number of input basis would be decreased down to the rank of the transmission matrix.

- (2) In the presence of measurement noise, their effects on the cross-correlations are not easily predictable in analytical approaches. We, therefore, performed numerical simulations to test the tolerance of our method in noisy situations (lines 125-138, and Fig. 4 in the revised manuscript).

We agree that the previous manuscript does not provide enough discussion about this issue, and we have accordingly revised the manuscript (Supplementary Note 1).

Similarly it would be useful and instructive to know how the results are expected to change if one increases either N or M . Is it worthwhile to maximise those numbers, or there is a trade off?

We thank the reviewer for this critical comment. When increasing either N or M , the following changes are expected:

1. Increasing N (\leq number of SLM pixels used)
 - In principle, increasing N is beneficial. Since the retrieved hologram is mapped on the N -preset basis, N represents the number of measurable information. Therefore, increasing N enables, for example, the FOV or NA of the holographic image sensor to be enhanced.
 - In practice, however, increasing N linearly extends the time for calibration of the transmission matrix, which will prevent practical implementations or applications.
2. Increasing M (\leq number of used camera pixels)
 - Similar to the case with N , increasing M is also beneficial. For a fixed N , increasing M/N suppresses the intrinsic noise originating from the second term in Eq. (3). Therefore, the retrieved hologram (x'_0) effectively converges to the incident field, as M increases. Furthermore, the modified Gerchberg-Saxton (GS) algorithm converges faster for a larger M/N .

However, in practical situations, setting proper N and M is an important issue, particularly when the computational power and memory and the acquisition speed are limited. If longer iteration time is not a concern, the choice of a system with the maximized N and the minimized M/N would provide the best holographic imaging quality. Since the optimal M/N value depends on the noise level of the

system (Supplementary Note 2), we recommend finding an appropriate M/N with a small N first and then maximizing N with the fixed M/N .

In the revised manuscript, a detailed discussion on this issue is included, and we believe this has enhanced the quality of the manuscript (lines 100-107).

The authors also never really discuss the fact that it is effectively impossible to measure the whole transmission matrix, and one always has to be happy with a subset of it. How big is the fraction of the matrix element the authors can measure? Does it make sense to push for a large fraction? Does the fact that correlations will appear if one measure a large fraction of the transmission matrix create a limit on the possible resolution of this method?

We thank the reviewer for bringing our attention to this point. Based on the working principle of our method, the following two conditions on a transmission matrix must be satisfied:

1. t_1, t_2, \dots, t_N , and y are randomly diffused (complex Gaussian distribution)
2. Orthogonality between the column vectors of transmission matrix, $\frac{1}{\sqrt{\Sigma_p \Sigma_q}} \langle t_p^* t_q \rangle_r = \delta_{pq}$

Since our method solely relies on the mathematical moment relations of chaotic fields [Eq. (2)], we expect any physical property of the diffuser (i.e. thickness, the degree of scattering, and absorption), or the fraction of the transmission matrix will not matter *unless* the above two conditions are violated. However, we would like to emphasize that the aspect ratio (M to N ratio) of the matrix (rather than the size of the matrix) is a crucial factor of the retrieved hologram quality (lines 125-138, Supplementary Fig. 2, and Fig. 4 in the revised manuscript). We have revised the manuscript accordingly.

Furthermore, what about stability? What happens is the diffuser, once characterized, is not perfectly aligned? How much tolerance does this method allows for? All these questions deserve some thought and discussion, especially because the authors are writing a paper on a novel technique that they hope will be used by others.

We thank the reviewer for the helpful comments. As the reviewer commented, the stability of this holographic lens system, which includes an aperture, a diffuser, a polarizer, and a camera, is very important. Since the output speckle field does not have any correlation along the lateral axes, slight changes such as translation/tilt of diffuser/camera induce significant decorrelation of the system, and it would not operate properly before the re-calibration of the transmission matrix.

To avoid this unwanted situation, we have designed the holographic image sensor so that components are firmly assembled in a lens tube. With the system we built, we successfully operated the system even one month after one-time calibration of the transmission matrix.

In the revised manuscript, we have included detailed information on this issue (lines 125-138, and Fig. 4).

Finally I would like to note that the presented technique is very computational in nature. Contrary to what fig. 1d suggests, this technique does not create a holographic image on a detector, but the only measured quantity is an apparent mess which can then be used to computationally recover the desired amplitude and phase. This is absolutely not a problem, but it would be nice to see it acknowledged in

the manuscript.

We thank the reviewer for bringing our attention to this point. Accepting the comment, we revised Figs. 1b and 1d to prevent the misunderstanding.

In conclusion this manuscript presents an intriguing and novel technique to measure the phase of an unknown wavefront, but leaves a large number of unanswered questions, that must be addressed before anyone can seriously think about implementing this method as a research tool.

We appreciate the kind compliments and helpful comments on our work. We have answered and revised the manuscript according to the comments above.

Reviewers' Comments:

Reviewer #2 (Remarks to the Author):

The authors have made substantial revisions. According to my suggestions, they included a detailed analysis of the tolerance of their method in practical (noisy) situations. Furthermore they provided a more detailed descriptions of the NA and the FOV.

I also appreciate that they toned down the claim regarding the future application and discussed the limitations their method.

In my opinion the revised version can be accepted for publication.

Reviewer #3 (Remarks to the Author):

The manuscript was significantly improved since the first submission, and now I am convinced it contains enough discussion on the experimental implementation to allow an interested researcher to reproduce the results.

One (minor) point: when, in the previous review round, I mentioned correlations in the transmission matrix I did not mean correlations in the "real" transmission matrix (which, as the authors correctly notice, are negligible in the diffusive regime). What I meant is that the "measured" transmission matrix often contains spurious correlations, mostly due to the fact that in practice it is difficult to really illuminate the sample with perfectly orthogonal wavefronts (see refs 22 and 23, but also arXiv:1511.04766 for examples of this happening). The solution is usually to filter the singular value spectrum to eliminate the modes with very small singular values (although the cut-off is sometimes a bit arbitrary). As far as I understand this is exactly what the authors do. If they are not doing it, then I suggest they explain how they solve the problem.

In conclusion I find the demonstrated technique to be novel, interesting, and well demonstrated. The manuscript is now at a suitable level of clarity, and the supplementary information contain enough details to make the reproduction of the results a viable task for any interested researcher. I strongly recommend publication.

Reviewer #2 (Remarks to the Author):

The authors have made substantial revisions. According to my suggestions, they included a detailed analysis of the tolerance of their method in practical (noisy) situations. Furthermore they provided a more detailed descriptions of the NA and the FOV. I also appreciate that they toned down the claim regarding the future application and discussed the limitations their method. In my opinion the revised version can be accepted for publication.

We appreciate the reviewer's comment.

Reviewer #3 (Remarks to the Author):

The manuscript was significantly improved since the first submission, and now I am convinced it contains enough discussion on the experimental implementation to allow an interested researcher to reproduce the results.

We appreciate the reviewer's comment.

One (minor) point: when, in the previous review round, I mentioned correlations in the transmission matrix I did not mean correlations in the "real" transmission matrix (which, as the authors correctly notice, are negligible in the diffusive regime). What I meant is that the "measured" transmission matrix often contains spurious correlations, mostly due to the fact that in practice it is difficult to really illuminate the sample with perfectly orthogonal wavefronts (see refs 22 and 23, but also arXiv:1511.04766 for examples of this happening). The solution is usually to filter the singular value spectrum to eliminate the modes with very small singular values (although the cut-off is sometimes a bit arbitrary). As far as I understand this is exactly what the authors do. If they are not doing it, then I suggest they explain how they solve the problem.

We thank the reviewer for careful reading. In a nutshell, we have done exactly the same way that the reviewer mentioned. The very small singular values means their corresponding speckle fields are very small, $\Sigma_p = \left\langle |t_p|^2 \right\rangle_r \cong 0$, and makes Eq. (1) unstable. In such a case, as the reviewer correctly commented, it is indeed necessary to filter out the small singular values, and effectively reduces the number of basis vectors (N). However, in our experimental situation, we did not observe the dimensional losses, and simply whole basis vectors have fully utilized without filtering. Reflecting the reviewer's comment, we appended a short discussion in Supplementary Note 1 about the situation when the transmission matrix is not a full rank.

In conclusion I find the demonstrated technique to be novel, interesting, and well demonstrated. The manuscript is now at a suitable level of clarity, and the supplementary information contain enough details to make the reproduction of the results a viable task for any interested researcher. I strongly recommend publication.

We appreciate the recommendation.